# Analysis of Bacterial and Metabolic Diversity of Pickles in Different Karst Regions of Guizhou, China

**DOI:** 10.3390/foods14081324

**Published:** 2025-04-11

**Authors:** Xiaokang Huang, Duhan Xu, Pan Wang, Cheng Chen, YuJia Wang, Yubo Zhang, Guangrou Lu, Mingjie Zhang, Ping Li, Chao Chen

**Affiliations:** College of Animal Science, Guizhou University, Guiyang 550025, China; h15329727872@163.com (X.H.); l8718227x1@163.com (D.X.); 18685204997@163.com (P.W.); cc376651938@163.com (C.C.); 17785844823@163.com (Y.W.); zyb0129gz@sina.cn (Y.Z.); l18185450732@163.com (G.L.); 18286640532@163.com (M.Z.)

**Keywords:** pickles, high-throughput sequencing, bacterial community, metabolites, karst regions

## Abstract

The unique geographical environment of karst regions provides pickles with a favorable flavor and taste; however, the contribution of the microbial community to pickle fermentation has not been fully explored. In this study, high-throughput sequencing and untargeted metabolomics were used to characterize 60 naturally fermented pickle samples from 12 different karst regions. The bacterial communities and metabolites of naturally fermented pickles changed significantly between different karst regions. *Lactobacillus delbrueckii*, *L. homohiochii*, and *L. fermentum* were the dominant bacterial species in pickle samples, with relative abundances of 29.66, 8.05, and 7.12%, respectively. There exist significant variations in the core biomarkers of traditional pickles across diverse regions characterized by rocky desertification and varying temperatures. Both *L. homohiochii* and *L. buchneri* stimulated complicated interspecies interactions in the bacterial community. *Lactobacillus* species exhibit excellent inhibitory effects against harmful bacterial populations under E-low- and E-high-temperature conditions. In total, 1976 metabolites were identified in pickles, including many previously undiscovered metabolites (e.g., Citrulline, GABA, tyrosol, and L-hydroxyroline) attributable to the dominances of *L. homohiochii*, *L. brevis*, *L. buchneri*, and *L. plantarum*. Lower levels of biogenic amines were found in pickles from the low-temperature regions. Furthermore, *L. delbrueckii* and *L. fermentum* were significantly negatively (*p* < 0.05) correlated with spermine and tyramine, and *Weissella cibaria* was negatively (*p* < 0.05) correlated with histamine. These data indicated that a low-temperature environment may be beneficial to the fermentation of pickles. This work provides new insights into the flavor of pickles resulting from the geological distribution of bacterial flora in karst regions.

## 1. Introduction

Pickles hold considerable economic importance due to their distinctive taste and nutritional composition, rendering them widely favored and cherished by consumers globally. Lactic acid bacteria (LABs) are commonly found in pickles owing to their capacity to generate substantial amounts of lactic acid and endure in highly acidic environments [1]. LABs are capable of generating diverse aroma compounds, bacteriocins, and exopolysaccharides, which play a crucial role in promoting the texture, flavor, and shelf life of fermented pickles [2]. Pickles exhibit a tangy and mild taste profile and are rich in essential nutrients such as vitamin C, vitamin B, organic acids (e.g., lactic acid, citric acid, fumaric acid, and malic acid), amino acids, alcohols, and esters, which promote digestive health. The intricate microbial consortia involved in pickle fermentation orchestrate a spectrum of biochemical processes that underlie the distinct taste of pickles. The principal metabolic pathways responsible for flavor compound synthesis encompass carbohydrate metabolism, fatty acid metabolism, and protein/amino acid breakdown [3,4].

Many factors can affect the fermentation process and by-products of pickles, including raw materials, microbiota, ambient temperature, and processing methods [5]. For instance, Northeast sauerkraut and pickles primarily undergo fermentation at low to moderate temperatures (5–15 °C), whereas Sichuan pickles are predominantly fermented at moderate to high temperatures (15–30 °C). Temperature affects pickle fermentation rate, consequently impacting maturation time [6]. The microorganisms and metabolites present in traditional fermented foods are intricately linked to the geographical location, climate, and temperature. However, previous research has predominantly centered around the microbial diversity and metabolic traits of traditional pickles, with limited attention given to the influence of varying geographical and climatic conditions on the bacterial community and metabolic composition of pickles.

Guizhou pickles are renowned for their distinctive sourness, spicy flavor, and crunchy consistency. Various vegetables, such as white-skinned radish, cabbage, cowpeas, and bamboo shoots, undergo pre-treatment and are immersed in a 6–8% (*w*/*v*) salt solution with spices and additives (including garlic, fresh chili, ginger, and vinegar). Subsequently, they are stored at room temperature (10–25 °C) for anaerobic fermentation by indigenous microorganisms present on the raw materials. Guizhou Province faces significant rocky desertification challenges in Southwest China due to factors such as high population density, economic underdevelopment, and a unique climate [7]. These issues contribute to soil degradation and alterations in microbial communities in the region [8]. The interplay among microbes, vegetation, and soil is crucial, and the diverse soil microbial communities in different areas result in variations in the microbial composition of the pickled raw materials. The karst regions in Guizhou, China, are widely representative in the world. The unique geographical environment of Guizhou’s karst regions may endow pickles with a distinctive flavor and nutritional value, and the abundant microbial resources within it require further exploration. Improving the quality of fermented vegetables may increase their nutritional value and provide health benefits. Therefore, it is necessary to understand the microbial community composition of pickles in different areas of Guizhou and its correlation with pickle metabolites, and to explore the geographical dependence of traditional pickles in Karst Areas.

High-throughput sequencing is an effective technique for analyzing the microbiomes of fermented foods. Metabolomics refers to the systematic identification and quantification of metabolites present in organisms, cells, and tissues. Therefore, the objective of this study was to analyze the bacterial communities and metabolites of pickles in different karst areas of Guizhou.

## 2. Materials and Methods

### 2.1. Pickle Sampling

Five homemade pickle samples were collected from 12 different regions in Guizhou Province, China. The different rocky desertification (LRD regions: Weining County, Huaxi District, Fenggang County, Renhuai County; MRD regions: Qixingguan District, Qianxi County, Yinjiang County, Libo County; IRD regions: Shuicheng County, Panzhou County, Guanling County, Wangmo County) (LRD, light rocky desertification; MRD, moderate rocky desertification; IRD, intense rocky desertification) and temperature regions (E-low-temperature regions, 8–12 °C: Weining County, Qixingguan District, Shuicheng County; low-temperature regions, 12–16 °C: Huaxi District, Qianxi County, Panzhou County; high-temperature regions, 16–20 °C: Fenggang County, Yinjiang County, Guanling County; E-high-temperature regions, 20–24 °C: Renhuai County, Libo County, Wangmo County) investigated in this study are shown in Figure 1. Samples were stored at −80 °C before analysis. Guizhou pickles are made by pretreating various vegetables such as white radish, cabbage, cowpeas, and bamboo shoots, soaking them in a 6–8% (*w*/*v*) salt solution containing spices and additives (including garlic, fresh chili, ginger, and vinegar), and then placing them at ambient temperature (10–25 °C) for anaerobic fermentation by local microorganisms. All raw materials were provided locally, and the production method was consistent.

### 2.2. PacBio Sequencing of Bacterial Communities

Total genome DNA from samples was extracted using SDS (Sodium Dodecyl Sulfate) method. DNA concentration and purity was monitored on 1% agarose gels. According to the concentration, DNA was diluted to 1 ng/μL using sterile water. 16S rRNA genes of distinct regions were amplified used specific primer with the barcode. All PCR reactions were carried out with TransStart^®^ FastPfu DNA Polymerase (TransGen Biotech, Beijing, China). Mix same volume of 1X loading buffer (contained SYB green) with PCR products and operate electrophoresis on 2% agarose gel for detection. PCR products was mixed in equidensity ratios. Then, mixture PCR products was purified with QIAquick@ Gel Extraction Kit (QIAGEN, Frankfurt, Germany). Sequencing libraries were generated using SMRTbellTM Template Prep Kit (PacBio, Menlo Park, CA, USA) following manufacturer’s recommendations. The library quality was assessed on the Qubit@ 2.0 Fluorometer (Thermo Scientific, Waltham, MA, USA) and FEMTO Pulse system. At last, the library was sequenced on the PacBio Sequel platform.

### 2.3. Ultra-High-Performance Liquid Chromatography (UHPLC)-Mass Spectrometry (MS/MS) Analysis of Pickles

Tissues (100 mg) were individually flash-frozen in liquid nitrogen and then ground, and the homogenate was resuspended in pre-chilled 80% methanol by vortexing. The samples were incubated on ice for 5 min and then centrifuged at 15,000× *g* for 20 min at 4 °C. Some of the supernatant was diluted to a final concentration of 53% in methanol by LC-MS-grade water. The samples were subsequently transferred to a fresh Eppendorf tube and then centrifuged at 15,000× *g* for 20 min at 4 °C. Finally, the supernatant was injected into an LC-MS/MS system for analysis [9]. UHPLC-MS/MS analyses were performed using a Vanquish UHPLC system (Thermo Fisher, Weil am Rhein, Germany) coupled with an Orbitrap Q Exactive^TM^ HF mass spectrometer or an Orbitrap Q Exactive^TM^ HF-X mass spectrometer (Thermo Fisher, Weil am Rhein, Germany) at Novogene Co., Ltd. (Beijing, China). Samples were injected onto a Hypersil Gold column (100 × 2.1 mm, 1.9 μm) using a 12 min linear gradient at a flow rate of 0.2 mL/min. Eluents A (0.1% FA in water) and B (methanol) were used for the positive polarity mode. The eluents used for the negative polarity mode were 5 mM ammonium acetate (eluent A, pH 9.0) and methanol (eluent B). The solvent gradient was set as follows: 2% B, 1.5 min; 2–85% B, 3 min; 85–100% B, 10 min; 100–2% B, 10.1 min; 2% B, 12 min. Q Exactive^TM^ HF mass spectrometer was operated in positive/negative polarity mode with spray voltage of 3.5 kV, capillary temperature of 320 °C, sheath gas flow rate of 35 psi, aux gas flow rate of 10 L/min, S-lens RF level of 60, and aux gas heater temperature of 350 °C. Raw data files generated by UHPLC-MS/MS were processed using Compound Discoverer (version 3.3; CD 3.3, Thermo Fisher, Weil am Rhein, Germany) to perform peak alignment, peak picking, and quantitation for each metabolite.

**Figure 1 foods-14-01324-f001:**
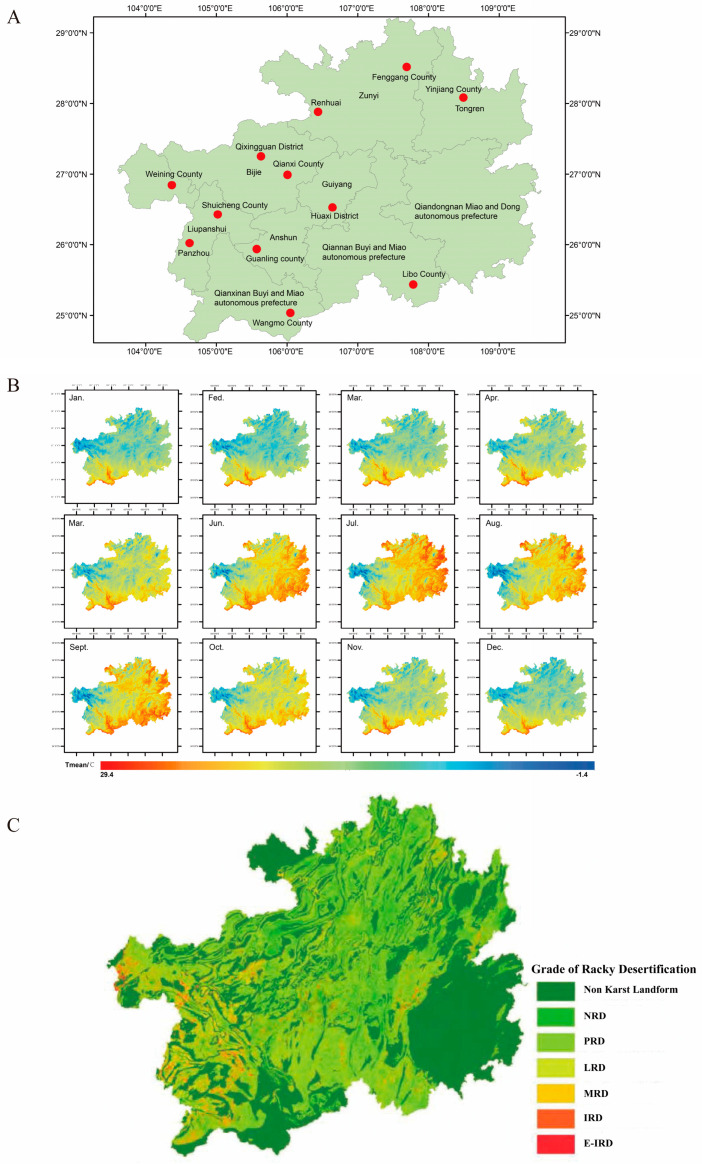
Geographical location (**A**) of sampling points, conditions (**B**), and grade of rocky desertification (**C**). Abbreviations: NRD, non-rocky desertification; PRD, potential rocky desertification; LRD, light rocky desertification; MRD, moderate rocky desertification; IRD, intense rocky desertification; E-IRD, extremely intense rocky desertification. Temperature data source: The 1 km monthly mean temperature dataset for China (1901–2023) [10]. National Tibetan Plateau Data Center (in Chinese). Source of rocky desertification data: Spatial Coupling Between Regional Green Urbanization and Rocky Desertification: A Case Study of Guizhou Province. Resources and environment in the Yangtze basin (in Chinese) [11].

### 2.4. Multivariate Statistical Analysis

The PacBio offline data were exported to a BAM format file. The Lima software platform (SMRT Link version 11.0) was used to differentiate the data of each sample according to barcode sequences, which saved all sample sequences in the BAM format. Subsequently, CCS (SMRT Link version 7.0) was employed for sequence correction, utilizing correction parameters of minfullpass = 3 and a minimum accuracy of 0.99 [12]. Sequences shorter than 1340 bp and longer than 1640 bp were eliminated and preserved in the fastq and fasta formats, respectively. Subsequently, SSR filtering was conducted, and cutadapt was employed to eliminate primers, filtering out sequences containing consecutive identical base pairs exceeding eight. Following these processes, final valid data (clean reads) were obtained. Using the Uparsene software (version 7.0.1001; http://drive5.com/uparse/, accessed on 8 January 2024), all clean reads of all samples were clustered into operational taxonomic units (OTUs) with a default consistency of 97%. According to the algorithm principles, sequences with the highest frequency of occurrence in OTUs were selected as representative OTU sequences. Species annotation of representative OTU sequences was conducted using the Mothur method and SILVA (http://www.arb-silva.de/, accessed on 8 January 2024) on the SSUrRNA database (with a threshold set at 0.8–1) [13]. Subsequently, taxonomic information was obtained, and the community composition of each sample at various taxonomic levels (kingdom, phylum, class, order, family, genus, and species) was statistically analyzed. The Chao1 and Shannon indices were calculated using the Qiime software program (version 1.9.1). aThe “vegan” package in the R software program (version 2.15.3) was used to perform a non-metric multidimensional scaling (NMDS) analysis. A linear discriminant analysis effect size (LEfSe; LDA Score > 4, *p*-value < 0.05, *q* < 0.05) was used to identify potential differences in microbial characteristics between the groups. Spearman’s coefficient between bacterial genera based on species abundance was calculated using Cytoscape (version 3.10.2) to construct network diagrams.

Metabolites were annotated using the Kyoto Encyclopedia of Genes and Genomes (KEGG) database (https://www.genome.jp/kegg/pathway.html, accessed on 6 November 2023). A partial least squares discriminant analysis (PLS-DA; *p*-value < 0.05, *q* < 0.05) was performed using MetaX for processing metabolomics data [14]. We applied a univariate analysis (*t*-test) to calculate the statistical significance (*p*-value). The metabolites with VIP > 1, *p*-value < 0.05, and fold change ≥ 1.5 were considered to be differential metabolites. For clustering heatmaps, the data were normalized using z-scores of the intensity areas of differential metabolites and plotted using the “pheatmap” package in R. Spearman’s correlation analysis was performed to determine the significant correlations between microbial communities and metabolites (*p* < 0.05).

## 3. Results and Discussion

### 3.1. Bacterial Community Composition of Pickles

Rarefaction curves of samples analysis can be viewed in the Appendix A. The bacterial diversity in pickles from different karst regions is shown in Figure 2. There were 568, 454, and 364 bacterial OTUs in the LRD, MRD, and IRD regions, respectively, and 327, 230, 470, and 476 bacterial OTUs in the E-low-, low-, high-, and E-high-temperature regions, respectively. Furthermore, pickles from the different rocky desertification or temperature regions shared 150 and 93 bacterial OTUs, respectively (Figure 2A,D). To understand the species distribution in the samples, Chao1 and Shannon indices were used to estimate species richness and bacterial diversity, respectively (Figure 2B,C,E,F). Higher bacterial alpha (α)-diversity was observed in samples from LRD and E-low-temperature regions. NMDS analysis showed that samples from different regions were located in different areas of the scoring diagram, indicating significant differences in their bacterial communities. The results indicated that the bacterial community composition of pickles varied according to region, with higher bacterial diversity and lower species richness observed in household pickles from the IRD regions. The bacterial communities of pickled raw materials in different rocky desertification areas are different, which may be due to the correlation between soil bacterial diversity and the level of rocky desertification [15]. Compared to high-temperature areas, the bacterial diversity and species richness of household pickles in low-temperature areas were significantly reduced, as bacterial community diversity increases with increasing fermentation temperature [16]. However, bacterial diversity and species richness in areas with the lowest temperatures were higher than those in areas with slightly low temperatures, which is likely a new discovery that urgently requires verification.

The bacterial community composition of pickles from different regions in Guizhou Province is shown in Figure 3. At the phylum level, *Firmicutes* (IRD > MRD > LRD; temperature, Low > High) and *Proteobacteria* (LRD > MRD >IRD; temperature, High > Low) were present, which is consistent with most previous studies [17,18]. In summary, the bacterial community composition of pickles varied in different regions of Guizhou Province. Pickles from areas with intense rocky desertification had higher abundances of *Firmicutes* and *Lactobacillus* and a lower abundance of *Proteobacteria* than those from areas with light to moderate rocky desertification. This may be due to the lower abundance of *Proteobacteria* in the soil microbial communities in areas with severe rocky desertification [19]. Pickles from low-temperature areas contained higher abundances of *Firmicutes* and *Lactobacillus* and a lower abundance of *Proteobacteria* than those from high-temperature areas because most *Lactobacillus* species are mesophilic [20]. With the intensification of rocky desertification and decrease in temperature, the abundance of *Lactobacillus delbrueckii* and *Lactobacillus fermentum* gradually increased, and the abundance of *Lactobacillus homohiochii* gradually decreased. In addition, an LEfSe analysis was used to identify biomarkers in pickles from different regions. As shown in Figure 4A,B, *Lactobacillus delbrueckii* and *Lactobacillus fermentum* were enriched in IRD regions; *Lactobacillus buchneri*, *Lactobacillus homohiochii*, *Ralstonia*, and *Stenorophomonasn* were determined as the biomarkers in MRD regions; and *Pediococcus* was identified as a biomarker genus in LRD regions. Regarding temperature, *Delftia* and *Lactobacillus buchneri* were enriched in E-low-temperature regions; *Lactobacillus plantarum* and *Stenorophomonasn* were the biomarkers in high-temperature regions; and *Lactobacillus homohiochii* and *Ralstonia* were biomarkers in E-high-temperature regions. These results indicate that the distribution of biomarkers is related to the regional environment, which is similar to that of Baijiu Daqu fermentation [21].

During fermentation, the microorganisms in pickles gradually form a relatively stable microbial community from a relatively unstable state through competition, symbiosis, and antagonism. To explore the relationship between the dominant bacterial communities in pickles from different karst regions, a network correlation analysis was conducted on the dominant genera in pickles from different karst regions based on Spearman’s correlation coefficient analysis (|r| > 0.6, *p* < 0.05; Figure 5). Network maps of LRD and MRD regions in different rocky desertification were divided into two modules. The nodes of the bacteria–bacteria co-occurrence networks in the LRD, MRD, and IRD regions were 42, 31, and 30, respectively. The number of edges in the LRD network was higher than that in the MRD and IRD regions, indicating that the network in the LRD region is more complex. In addition, except for links connected to *Lactobacillus*, which had negative effects [20], all other network connections had positive effects, and there were more negative effects in MRD regions, indicating that *Lactobacillus* is the best at inhibiting other harmful bacterial populations in MRD regions. In the different temperature regions, the nodes of the bacteria–bacteria co-occurrence networks in the E-low-, low-, high-, and E-high-temperature regions were 34, 26, 35, and 40, respectively. The number of edges in the E-high-temperature network was higher than that in the E-low-, low-, and high-temperature regions, indicating that the network in the E-high-temperature regions was more complex. Furthermore, the networks of low- and high-temperature regions had positive effects, but in the networks of the E-low- and E-high-temperature regions, *Lactobacillus* had a particularly negative effect. This may be related to the regions and microenvironments in which they reside. An investigation of biotic interactions and the subsequent effects resulting from the presence or absence of specific microorganisms is of utmost importance. Other studies have also shown that microbial interactions can drive community succession, and dominant genera play different key roles in fermentation [22]. In summary, during pickle fermentation, *Lactobacillus* can effectively inhibit many harmful bacterial genera, such as *Delfia*, *Acinetobacter*, *Alcaligenes*, *Ralstonia*, *Stenotrophomonas*, and *Enterbacter*. Moreover, *Lactobacillus* exhibited excellent inhibitory effects on harmful bacterial communities under E-low- and E-high-temperature conditions, compared with typical low or high temperatures.

### 3.2. Metabolites of Pickles

QC (quality control) sample correlation analysis and MS/MS raw data and match scores for metabolites can be viewed in the Appendix A. In this study, a total of 1976 metabolites of pickles were identified, which were classified into 14 categories based on their chemical structures, including lipids and lipid-like molecules (29.10%), phenylpropanoids and polyketides (14.93%), organic acids and derivatives (14.37%), organoheterocyclic compounds (14.37%), benzenoids (9.67%), organic oxygen compounds (7.64%), nucleosides, nucleotides, and analogs (4.00%), alkaloids and derivatives (3.04%), organic nitrogen compounds (1.57%), lignans, neolignans and related compounds (1.06%), organosulfur compounds (0.10%), mixed metal/non-metal compounds (0.05%), homogeneous non-metal compounds (0.05%), and hydrocarbons (0.05%) (Figure 6A). According to the KEGG database annotation, pickle metabolites were mainly divided into four functional categories: cellular processes (0.19%), environmental information processing (4.00%), genetic information processing (1.77%), and metabolism (94.04%). Based on the metabolic processes involved, the 1074 metabolites were further divided into 16 KEGG pathways. Overall, 2 metabolites are involved in transport and catabolism, 10 in signal transduction, 33 in membrane transport, 16 in translation, 3 in folding, sorting, and degradation, 56 in nucleotide metabolism, 32 in the metabolism of terpenoids and polyketides, 36 in the metabolism of other amino acids, 66 in the metabolism of cofactors and vitamins, 67 in lipid metabolism, 1 in glycan biosynthesis and metabolism, 376 in global and overview maps, 18 in energy metabolism, 65 in carbohydrate metabolism, 133 in biosynthesis of other secondary metabolites, and 160 in amino acid metabolism (Figure 6B). These results were similar to those of previous studies [17,23]. PLS-DA was used to explore the correlations between the samples to determine the overall distribution of pickle samples from different regions. There were significant differences in the metabolites of pickles from different regions in Guizhou Province, as shown in Figure 6C,D. This is because the microbial communities and metabolites of fermented foods are geographically dependent, and different fermentation environments can form different fermentation systems [24].

Different regions, fermentation materials, and microbial community structures enrich different metabolic pathways [25]. To better understand the differences in the metabolic pathways of pickles from different regions of Guizhou Province, we conducted metabolite expression pattern clustering and cluster functional enrichment analysis using the Mfuzz algorithm (Figure 7). The metabolites were divided into six and eight clusters based on their trend similarities in different rocky desertification and temperature regions, respectively (Figure 7A,B). In different rocky desertification regions, the metabolites in cluster 1 were mainly detected in the purine metabolism and pyrimidine metabolism pathways; metabolites in cluster 2 were mainly detected in the histidine metabolism and nicotinate and nicotinamide metabolism pathways; metabolites in cluster 3 were mainly detected in the alanine, aspartate and glutamate metabolism, glycine, serine and threonine metabolism, linoleic acid metabolism, riboflavin metabolism, and pentose phosphate pathways; metabolites in cluster 4 were mainly detected in the arginine and proline metabolism, arginine biosynthesis, phenylalanine metabolism, phenylalanine, tyrosine and tryptophan biosynthesis, steroid hormone biosynthesis, and vitamin B6 metabolism pathways; metabolites in cluster 5 were mainly detected in the citrate cycle, cysteine and methionine metabolism, and glutathione metabolism pathways; and metabolites in cluster 6 were mainly detected in the arginine and proline metabolism, arginine biosynthesis, steroid hormone biosynthesis, and tyrosine metabolism pathway (Figure 7C). Clusters 1 and 5 had the highest expression levels in LRD regions, clusters 2 and 3 had the highest expression levels in IRD regions, and clusters 4 and 6 had the highest expression levels in MRD regions (Figure 7A). In different temperature regions, the metabolites in cluster 1 were mainly detected in the alanine, aspartate and glutamate metabolism, arginine biosynthesis, citrate cycle, galactose metabolism, purine metabolism, pyrimidine metabolism, and pentose phosphate pathways; metabolites in cluster 2 were mainly detected in the glyoxylate and dicarboxylate metabolism and steroid hormone biosynthesis pathways; metabolites in cluster 3 were mainly detected in the cysteine and methionine metabolism, glycine, serine and threonine metabolism, nicotinate and nicotinamide metabolism, and purine metabolism pathways; metabolites in cluster 4 were mainly detected in the phenylalanine metabolism pathways; metabolites in cluster 5 were mainly detected in the caffeine metabolism pathways; metabolites in cluster 6 were mainly detected in the amino sugar and nucleotide sugar metabolism, lysine degradation, and pantothenate and CoA biosynthesis pathways; metabolites in cluster 7 were mainly detected in the linoleic acid metabolism, tryptophan metabolism, and tyrosine metabolism pathways; and metabolites in cluster 8 were mainly detected in the arginine and proline metabolism, glutathione metabolism, tyrosine metabolism, and vitamin B6 metabolism pathway (Figure 7D). Clusters 1 and 4 had the highest expression levels in the high-temperature regions, clusters 2 and 5 had the highest expression levels in the E-low-temperature regions, clusters 3 and 6 had the highest expression levels in the low-temperature regions, and clusters 7 and 8 had the highest expression levels in the E-high-temperature regions (Figure 7B). In essence, varying rocky desertification levels and temperature zones give rise to distinct fermentation conditions (temperature), substrates for fermentation, and diversity and composition of microbial communities, consequently leading to variations in the enrichment of metabolic pathways across different fermentation processes.

Previous studies have shown that the same metabolites simultaneously participate in multiple metabolic pathways, indicating that the different metabolites have significant effects on the pathways [15]. Based on the differential metabolic pathways mentioned above, we identified the differential metabolites in pickles from different regions, as shown in Figure 8 (*p* < 0.05, VIP > 1, FC > 1.5). The differentially expressed metabolites were mainly carboxylic acids and their derivatives, fatty acyls, organonitrogen compounds, organooxygen compounds, and phenols. The expression levels of carboxylic acids and their derivatives in pickles from the IRD regions were significantly lower than those in pickles from the MRD and LRD regions (Figure 8A). The expression level of phenols in pickles from extremely high-temperature regions was higher than that in pickles from other temperature regions (Figure 8B). Metabolites and volatile compounds are mainly produced by microbial metabolism during fermentation [26], and the diversity of the pickle microbiota leads to differences in flavor. For a more intuitive display, we used Sankey plots to analyze the changes in various differential metabolites in pickles from different regions, as shown in Figure 9. In different rocky desertification regions, citrulline, L-arginine, citric acid, L-tyrosine, L-aspartic acid, Υ-aminobutyric acid (GABA), and 2-oxoglutaric acid have the highest expression levels in LRD regions and extremely low expression levels in MRD and IRD regions. 2-oxoglutaric acid and citric acid are the main intermediates in the citrate cycle pathway, and their expression levels were consistent with the highest citrate cycle pathway found in the LRD regions. In pickles, glutamate provides an umami taste, whereas tyrosine and arginine provide a bitter taste [27]. Citrulline, essential for arginine biosynthesis, is a potential therapeutic agent utilized in the management of conditions characterized by arginine deficiency [28]. GABA is produced through the decarboxylation of glutamate, and consumer demand for functional foods enriched with GABA is high due to its diverse health-promoting properties such as neuroprotection, anti-insomnia, antidepressant, antihypertensive, antidiabetic, and anti-inflammatory effects [29]. The expression levels of tyrosol, vanillyl mandelic acid, 4-guanidinobutyric acid, guanidineacetic acid, L-phenylalanine, L-serine, N-acetyl-Asp-Glu, L-hydroxyproline, spermidine, choline, gluconolactone, adenylosuccinic acid, 6-phosphogluconic acid, pyridoxine, and tyramine gradually decreased as the rocky desertification intensified. Tyrosol and vanillyl mandelic acid are metabolites of the tyrosine pathway. Tyrosol, a natural antioxidant, exhibits cytoprotective properties against oxidative stress, thereby conferring health benefits [30]. 4-Guanidinobutyric acid, guanidineacetic acid, and L-hydroxyproline are products of the alanine, aspartate, and glutamate pathways. 4-Guanidinobutyric acid exerts inhibitory effects on gastric disease progression [31]; guanidineacetic acid, a direct endogenous precursor of creatine, promotes skeletal muscle growth [32]; and L-hydroxyproline, a non-protein amino acid derived from proline hydroxylation, acts as a scavenger of reactive oxygen species [33]. N-acetyl-Asp-Glu and adenylosuccinic acid are metabolites of the alanine, aspartate, and glutamate pathways. In pickles, serine provides a sweet taste and phenylalanine provides a bitter taste [27]. Gluconolactone and 6-phosphogluconic acid are the main intermediates in the pentose phosphate pathway, and their expression levels are consistent with the lowest citrate cycle pathway found in the LRD regions. Pyridoxine is an essential nutrient for humans that is widely present in meat, whole grains, and vegetables [34]. Sauerkraut, produced by the fermentation of cabbage, is a good source of minerals and vitamins, and contains histamine, tyramine, putrescine, cadaverine, spermine, and spermidine [35]. Biogenic amines are produced by bacteria through decarboxylation of the corresponding amino acids in food. The concentration of biogenic amines in fermented foods is influenced by several factors during the production process, including raw material hygiene, microbial composition, fermentation conditions, and fermentation time. The ingestion of small amounts of biogenic amines does not usually have harmful effects on human health. However, when their levels in food are too high and their detoxification ability is inhibited or disrupted, biogenic amines can cause problems [36]. For example, histamine and tyramine in foods exceeding 100 mg/kg each can cause human poisoning [37]. The expression levels of fumaric acid, succinic acid, succinic semialdehyde, vitamin B2, S-adenosylmethionine, nicotinamide, palmitic acid, and histamine were highest in the LRD regions and lowest in the MRD regions. Fumaric acid, succinic acid, and succinic semialdehyde are mainly involved in the alanine, aspartate, and glutamate metabolism pathways. S-adenosylmethionine is required and of fundamental importance for the metabolism of nucleic acids and polyamines, the structure and function of membranes, and as a precursor of glutathione [38]. Nicotinamide is an amide compound of niacin that plays a role in protein and sugar metabolism and can improve nutrition in humans and animals. Palmitic acid is the most common saturated fatty acid accounting for 20–30% of the total fatty acids in the human body and can be provided in the diet [39]. Histamine is produced from histidine under the action of decarboxylase. Notably, the expression levels of ribulose-5-phosphate and D-erythrose 4-phosphate gradually increased as rocky desertification intensified, and spermine was almost only detected in the MRD regions. Effective measures should be taken to suppress spermine content in MRD regions. In different temperature regions, the expression of 4-aminobutyric acid, pantetheine, pipecolic acid, L-saccharopine, citric acid, L-serine, glucosamine, gluconolactone, inositol, dulcitol, homogentisic acid, 6-phosphogluconic acid, pyridoxine, and succinic semialdehyde gradually decreased with decreasing temperature. Pipecolic acid and L-saccharopine are products of the Lysine degradation pathway, and their expression levels were consistent with the higher lysine degradation pathway activity found in the E-low- and low-temperature regions. Inositol and dulcitol are products of the galactose metabolism pathway and growth factors for animals and microorganisms. The differences in their expression levels may be caused by the unique bacterial communities that utilize them. The expression of citrulline, L-arginine, L-aspartic acid, nicotinamide, 3-(2-Hydroxyethyl) indole, and 2-oxoglutaric acid was higher in the low-temperature regions than in the high-temperature regions, whereas the expression of tyrosine and spermine was higher in the high-temperature regions than in the low-temperature regions. Interestingly, the expression of 4-guanidinobutyric acid, 3,4-dihydroxyphenylglycol, vanillyl mandelic acid, and L-kynurenine was higher in areas with extremely low and high temperatures. However, tyrosol was almost exclusively present in pickles from extremely high-temperature regions, whereas fumaric acid showed higher expression levels in both extremely high- and low-temperature regions. In summary, there were significant differences in the metabolites of pickles in different regions in Guizhou Province, and the expression levels of different metabolites varied according to the regions. Guizhou pickles contain numerous bacteria and metabolites that are beneficial to human health. Notably, in the MRD and extremely high-temperature regions, the expression level of biogenic amines was higher than that in other regions, which may be related to specific local microorganisms. Local residents should take effective measures to improve the fermentation quality of pickles, such as storage under low-temperature conditions. Many factors can affect the pickle fermentation process, including raw materials, microbiota, ambient temperature, and processing methods [5]. In this study, we identified many previously undiscovered pickle metabolites, which may be due to the unique karst geological environment in Guizhou.

### 3.3. Combined Analysis of Bacterial Community and Metabolites in Pickles

Given that fermentation is based on microbial metabolism, microorganisms are closely related to fermentation metabolites [40]. To further explore the relationship between the bacterial community and the differential metabolites, based on Pearson’s algorithm, a correlation analysis of 15 core species and 44 differential metabolites was performed. The R software package was used for a correlation heatmap analysis (Figure 10A), and Cytoscape was used to perform a network analysis (*p* < 0.05, |r| > 0.25) (Figure 10B). In microbiome research, r > 0.2 and *p* < 0.05 are commonly regarded as significant signals. A total of 48 nodes and 102 edges (41 positive correlations and 61 negative correlations) were obtained. Among these bacteria, *Lactobacillus* plays an important role in the production of aroma, flavors, organic acids, and fatty acids in pickles [41]. In this study, *Lactobacillus delbrueckii* and *Lactobacillus fermentum*, which are the core bacteria of Guizhou pickles and biomarkers of IRD regions, showed significant positive correlations with pantetheine, vitamin B2, palmitic acid, succinic acid, succinic semialdehyde, pipecolic acid, nicotinamide, and L-saccharopine, and significant negative correlations with citrulline, fumaric acid, homogentisic acid, tyrosol, tyramine, L-phenylalanine, L-tyrosine, L-arginine, spermine, and guanidineacetic acid. *Lactobacillus homohiochii*, as a biomarker of the MRD and E-high-temperature regions, was positively correlated with tyramine, L-phenylalanine, and GABA, and negatively correlated with fumaric acid, pantetheine, nicotinamide, L-saccharopine, D-erythrose 4-phosphate, 6-phosphogluconic acid, and gluconolactone. *Lactobacillus buchneri*, as a biomarker of the MRD and E-low-temperature regions, was positively correlated with citrulline and spermine levels. *Lactobacillus plantarum*, the core bacteria of Guizhou pickles and biomarkers of high-temperature regions, showed significant positive correlations with L-tyrosine, L-arginine, L-hydroxyproline, and L-kynurenine and significant negative correlations with succinic acid, pantetheine, adenylosuccinic acid, vitamin B2, palmitic acid, and L-serine. Notably, *Lactobacillus* sp. *Marseille-P3825* was strongly positively associated with fumaric acid, pipecolic acid, nicotinamide, and S-adenosylmethionine. *Lactobacillus brevis* is associated with tyrosol, tyramine, spermine, L-kynurenine, and 2-oxoglutaric acid showed a positive correlation, which explains why tyrosol is almost exclusively found in pickles from extreme temperature regions. *Weissella cibaria* has been found in tropical fermented vegetables [42], and in this study, *Weissella cibaria* was found in IRD and high-temperature regions, which is negatively correlated with succinic acid, histamine, and S-adenosylmethionine. Interestingly, *Weissella paramesenteroides* showed a particularly strong negative effect on tyrosol (|r| = 0.97, *p* < 0.005), which has also been observed in various fermented foods. In summary, the relationship between the core microbiota and metabolites provides a good explanation for the changes in the differential metabolites in pickles from different regions. Single metabolites or microorganisms were associated with multiple microorganisms. Overall, there was a strong correlation between the microbiome and metabolites.

## 4. Conclusions

In this study, the microbial community and metabolites of naturally fermented pickles were shown to be geographically dependent in karst regions. *L. delbrueckii*, *L. homohiochii*, and *L. fermentum* were the dominant bacterial species in pickle samples, with relative abundances of 29.66, 8.05, and 7.12%, respectively. The genera of *Lactobacillus* exhibit excellent inhibitory effects on harmful bacterial communities at both E-low- (relative abundance: 78.14%) and E-high-temperatures (relative abundance: 66.93%). In total, 1976 metabolites were identified in pickles, including many previously undiscovered metabolites (e.g., Citrulline, GABA, tyrosol, and L-hydroxyroline). Moreover, low-temperature (8–16 °C) preservation can effectively inhibit biogenic amine synthesis in pickles attributable to the dominances of *L. delbrueckii* and *L. fermentum*. Therefore, we predicted that low-temperature preservation would improve the fermentation quality of pickles. This study not only analyzed the differences in bacterial communities and metabolites of naturally fermented pickles from various karst regions, but also provided new insights and theoretical support for the development and utilization of these pickles. But these findings are specific to karst regions, and validation in non-karst ecosystems is needed, and the exact antimicrobial compounds from *Lactobacillus* require isolation and structural identification.

## Figures and Tables

**Figure 2 foods-14-01324-f002:**
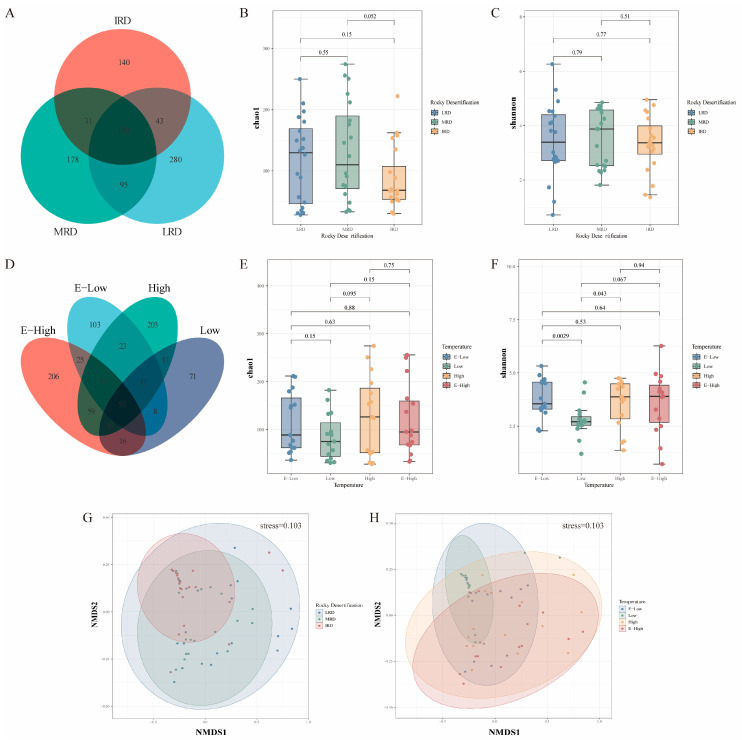
Differences in alpha and beta diversity indices of bacterial communities in pickles from different regions. OTUs (**A**), Chao1 (**B**), and Shannon (**C**) alpha diversity indices of bacterial communities in pickles from different rocky desertification regions. OTUs (**D**), Chao1 (**E**), and Shannon (**F**) alpha diversity indices of bacterial communities in pickles from different temperature regions. An NMDS analysis is used to compare the bacterial community of pickles from different regions ((**G**) different rocky desertification regions; (**H**) different temperature regions). Using an NMDS analysis, according to the species information contained in the sample, it is reflected in the multidimensional space in the form of points, and the degree of difference between different samples is reflected through the distance between points, which can reflect the differences between and within groups of samples. Abbreviations: LRD, light rocky desertification; MRD, moderate rocky desertification; IRD, intense rocky desertification; E-low, extremely low temperature; Low, low temperature; High, high temperature; E-high, extremely high temperature.

**Figure 3 foods-14-01324-f003:**
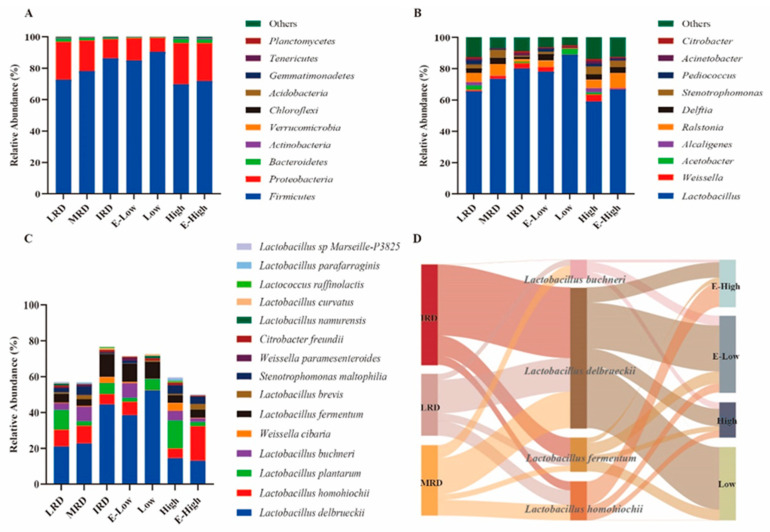
Relative abundance of bacteria community in pickle samples at phylum (**A**), genus (**B**), and species (**C**) levels including Lactobacillus delbrueckii, Lactobacillus fermentum, Lactobacillus homohiochii, and Lactobacillus buchneri ((**D**) *p* < 0.05) changes in different regions.

**Figure 4 foods-14-01324-f004:**
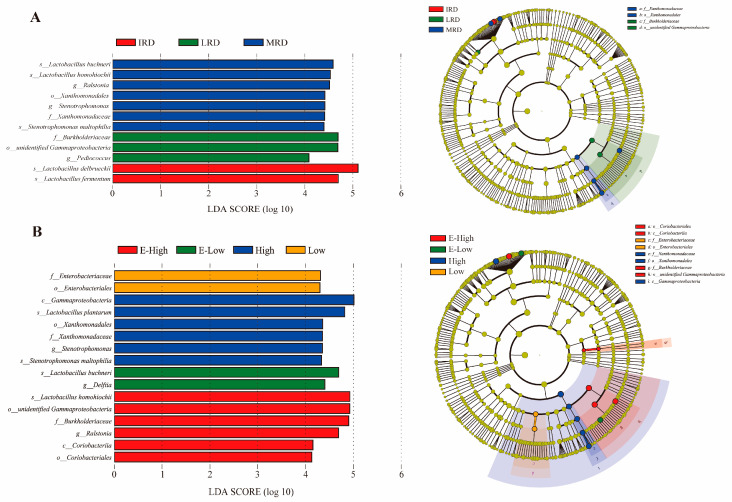
LEfSe analysis highlights differential bacteria in pickles from different regions ((**A**) different rocky desertification regions; (**B)** different temperature regions).

**Figure 5 foods-14-01324-f005:**
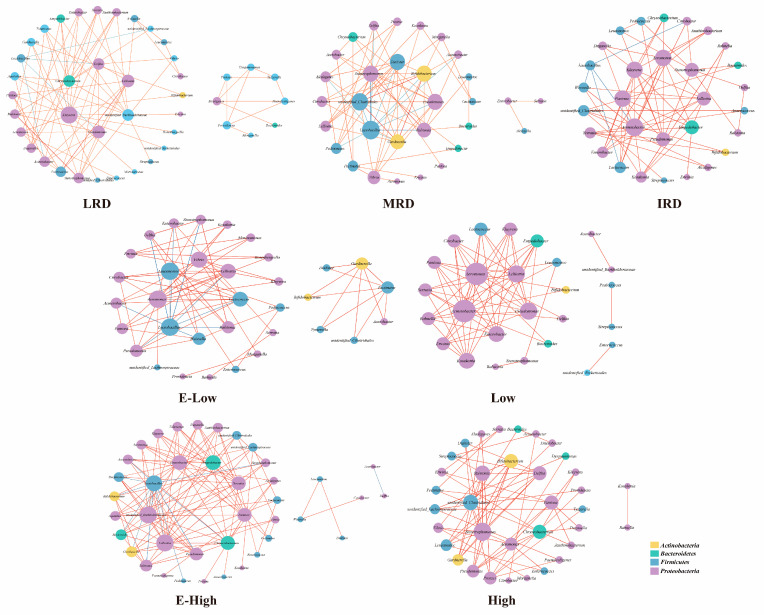
Network co-occurrence among genus level in pickles from different regions. Different nodes represent different genus level, nodes in the same phylum level have the same color, and node size was determined based on degree centrality. The color intensity of the connections between nodes is positively correlated with the absolute value of the correlation coefficient between species interactions, with red indicating a positive correlation and blue indicating a negative correlation.

**Figure 6 foods-14-01324-f006:**
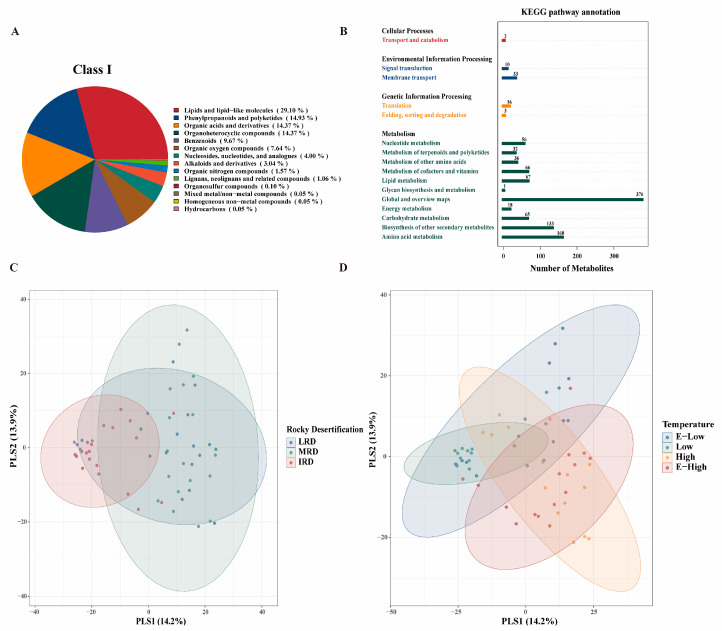
The primary classification of metabolites in pickles (**A**), KEGG pathway annotation (**B**), and PLSD analysis are used to compare the metabolites of pickles from different regions ((**C**) different rocky desertification regions; (**D**) different temperature regions).

**Figure 7 foods-14-01324-f007:**
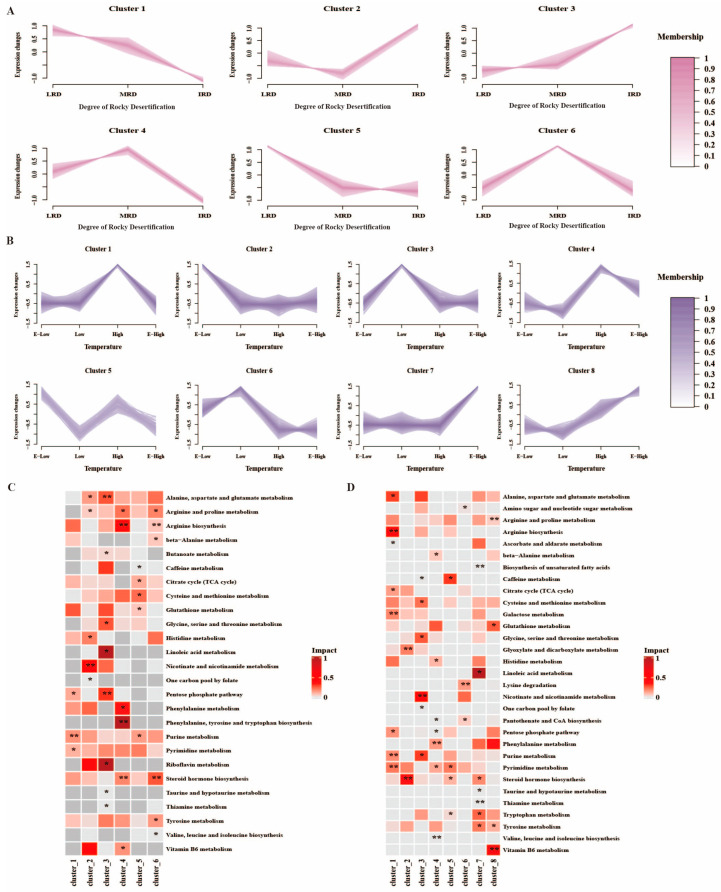
Metabolite expression pattern clustering ((**A**) different rocky desertification regions; (**B**) different temperature regions) and cluster functional enrichment analysis ((**C**) different rocky desertification regions; (**D**) different temperature regions) based on the Mfuzz algorithm (*: *p* ≤ 0.05; **: *p* ≤ 0.01).

**Figure 8 foods-14-01324-f008:**
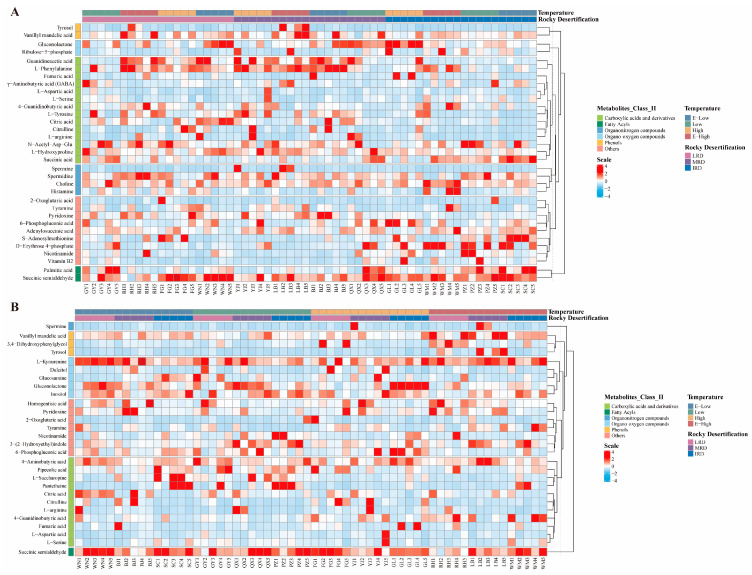
Cluster heatmap of metabolite cluster analysis among metabolic sets from different regions ((**A**) different rocky desertification regions; (**B**) different temperature regions). Each column in the figure represents a sample and each row represents a metabolite. The color represents the relative expression amount of metabolites in the group of samples.

**Figure 9 foods-14-01324-f009:**
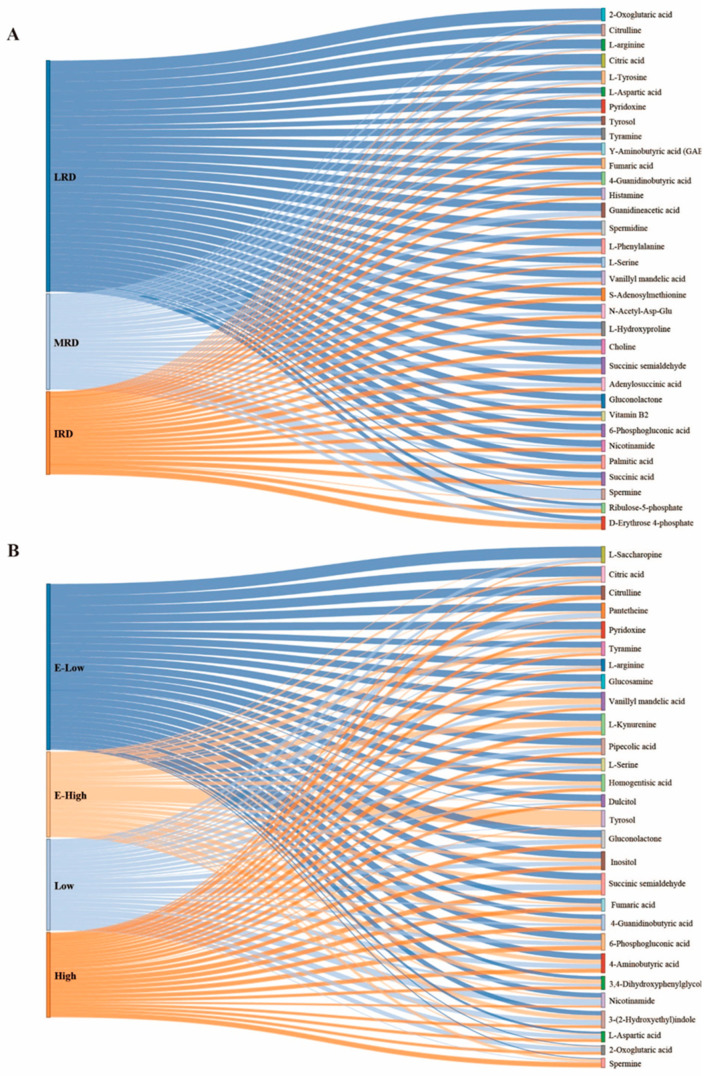
Sankey plots to analyze the changes in various differential metabolites of pickles from different regions ((**A**) different rocky desertification regions; (**B**) different temperature regions). The rectangle’s height reflects metabolite expression levels, with distinct colors assigned to each group and metabolite. Lines depict metabolite variations between groups, with line thickness indicating expression levels. Branch widths in the diagram correspond to data traffic volume, with the total width of all branches equaling the sum of all branched widths.

**Figure 10 foods-14-01324-f010:**
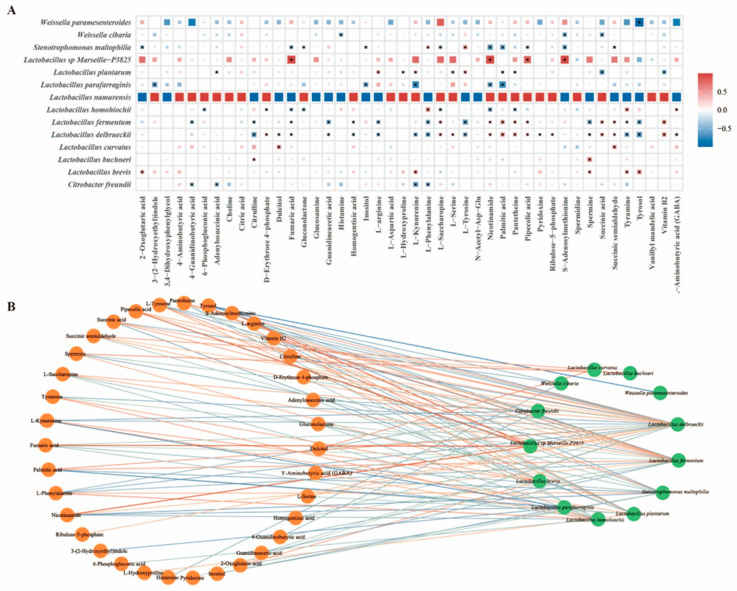
Heatmap analysis ((**A**) *: *p* ≤ 0.05) and network analysis ((**B**) *p* ≤ 0.05) of correlation between differential microorganisms and metabolites of pickles. Correlation coefficient is normalized and color intensity is proportional to the relevance, with red indicating a positive correlation and blue indicating a negative correlation.

## Data Availability

Where data is unavailable due to privacy or ethical restrictions.

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
