# Peer review of "Analysis of Bacterial and Metabolic Diversity of Pickles in Different Karst Regions of Guizhou, China"

_foods, 2025, doi:10.3390/foods14081324_

Round 1

Reviewer 1 Report

Comments and Suggestions for Authors

General Comments

This manuscript presents an interesting and timely investigation of the bacterial and metabolic diversity in pickles fermented in various karst regions of Guizhou, China. The use of high-throughput sequencing and untargeted metabolomics is appropriate and offers comprehensive insight into how environmental factors, such as temperature and rocky desertification, shape the microbial communities and metabolites in pickles. The topic is of potential interest to the readers of Foods and contributes to the growing body of literature exploring the terroir of fermented foods. However, the manuscript currently suffers from several critical weaknesses that limit its scientific clarity, reproducibility, and impact. These issues include poor organization, excessive redundancy, vague methodological descriptions, missing or poorly justified statistical approaches, lack of mechanistic depth, and weak discussion of the novelty. Therefore, major revisions are required before this manuscript can be considered for publication.

Concerns

#1 The abstract is too dense, overly detailed, and fails to effectively summarize the key findings with clarity and conciseness. Please add more quantitative information.

#2 – The manuscript requires thorough language editing by a native or professional editor, as awkward phrasing and grammatical errors impede readability.

#3 The authors claim novelty, but do not clearly state how this study advances previous work or differs from prior studies of microbiota in pickles or other regional fermented foods.

#4 More explanation is needed about how karst topography and rocky desertification affect microbial or soil characteristics relevant to fermentation.

#5 The choice of sampling sites is insufficiently justified. How were these 12 regions selected? Are they representative?

#6 Read depth, rarefaction curves, or control checks for contamination are not presented.

#7 The description of multivariate statistics (e.g., PLS-DA, LEfSe) is brief and does not explain whether corrections for multiple comparisons were applied.

#8 The use of LEfSe alone is not sufficient for biomarker validation. No independent validation of the microbial markers is presented.

#9 For example, stating that low temperatures “facilitate fermentation” without sensory or kinetic validation is unfounded.

#10 A major omission is the lack of sensory evaluation to link metabolites or bacteria with flavor/taste attributes.

#11 Some figures (e.g., NMDS, network plots) are overly complex and poorly labeled, lacking proper interpretation.

#12 The metabolomics workflow lacks detail about how technical variability was controlled.

#13 At times, species names are misspelled (e.g., Lactobacillus homohiochii vs homohaichii) and not properly italicized.

#14 The metabolite-microbiota correlations are not statistically strong or biologically contextualized. Most r-values are modest and should be interpreted cautiously.

#15 No non-karst controls or time-course data are provided to establish fermentation dynamics.

#16 While the manuscript mentions amines like spermine, there is no discussion on their regulatory thresholds or health impacts.

#17 Several figures, especially Sankey plots and network diagrams, are hard to interpret and require simplification or enhanced annotation.

#18 The manuscript reads more like a data dump than a coherent scientific narrative. The results lack a clear focus or prioritization.

#19 Untargeted metabolomics often requires MS/MS validation, which was not sufficiently discussed or shown.

#20 The authors summarize data but rarely interpret the biological significance or contrast their results with other studies.

#21 A more concise and data-driven conclusion is needed, with an emphasis on the actual contribution and limitations.

#22 A similarity rate above 25% in an original article raises serious concerns regarding excessive reliance on previously published materials. The authors must revise or paraphrase heavily matched sections to ensure originality. The similarity comes from a variety of academic and internet sources, including Frontiers, MDPI, Springer, and Food Chemistry. This indicates patchwriting or re-use of sentences from multiple references, rather than independent scientific writing. Several technical descriptions (especially in the Introduction, Materials and Methods, and Discussion) closely resemble content from previously published works. These should be rephrased in the authors’ own words.

Comments on the Quality of English Language

Please see my report!

Reviewer 2 Report

Comments and Suggestions for Authors

This study highlights the bacterial and metabolite diversity of traditional pickle, which is may add some novelty in the use of metagenomic and metabolomic technology in food science. However, the authors need to clarify in the Introduction, why do you need to understand the bacterial succession (correlated with certain metabolites) during fermentation. Is there any issues during fermentation, so that the authors interested to reveal the succesion? I guess this pickle is naturally fermented.

please add some references about NGS and metabolomic in the introduction.

please improve the quality of the figures, make it in high definition if possible.

Round 2

Reviewer 1 Report

Comments and Suggestions for Authors

The authors have revised their manuscript acordingly. Accept!